# Cutting-Edge Strategies for Renal Tumour-like Lesions in Granulomatosis with Polyangiitis: A Systematic Review

**DOI:** 10.3390/diagnostics14050566

**Published:** 2024-03-06

**Authors:** Luca Iorio, Marco Pizzi, Diego Cecchin, Federica Davanzo, Anna Ghirardello, Angelo Paolo Dei Tos, Andrea Doria, Roberto Padoan

**Affiliations:** 1Rheumatology Unit, Department of Medicine DIMED, University of Padua, 35121 Padua, Italyadoria@unipd.it (A.D.); 2Surgical Pathology and Cytopathology Unit, Department of Medicine DIMED, University of Padua, 35121 Padua, Italy; 3Nuclear Medicine Unit, Department of Medicine DIMED, University of Padua, 35121 Padua, Italy

**Keywords:** ANCA vasculitis, granulomatosis with polyangiitis, Wegener granulomatosis, renal tumour-like lesions, renal masses, pseudotumour, granuloma

## Abstract

Background: Granulomatosis with polyangiitis (GPA) is characterised by granulomatous inflammation and small-to-medium vessel necrotising vasculitis, mainly affecting respiratory tract and kidneys. Renal involvement presenting as tumour-like lesions poses diagnostic and treatment challenges. Methods: Following the observation of a GPA patient presenting with multiple renal tumour-like lesions, we conducted a systematic literature review on MEDLINE/PubMed, EMBASE, and Cochrane databases. Data gathered from the literature were analysed to summarise the diagnostic approach, management, and outcome of renal GPA-related tumour-like lesions. Results: a 49-year-old female presented with persistent constitutional symptoms and multiple bilateral renal lesions. Renal biopsy showed chronic interstitial inflammation with necrotising granulomas. Laboratory tests disclosed positive anti-proteinase 3 (PR3) anti-neutrophil cytoplasmic antibody (ANCA) leading to a final diagnosis of GPA. She was effectively treated with high-dose glucocorticoids and rituximab. Literature search yielded 41 articles, concerning 42 GPA patients with renal masses, presenting bilaterally in 23.8% of the cases. Positive PR3-ANCA was observed in 86.5% of the cases. Half of 42 patients showed kidney abnormalities. Treatment with glucocorticoids (83.3%) and immunosuppressive agents (80.9%) resulted in an overall good remission rate and favourable prognosis. Conclusions: GPA should be considered in the differential diagnoses of kidney tumour-like lesions. The diagnosis is challenging, and histological examination greatly contributes to the diagnostic work-up.

## 1. Introduction

Granulomatosis with polyangiitis (GPA), formerly known as Wegener’s granulomatosis, is a systemic autoimmune disorder characterised by granulomatous inflammation and small-to-medium vessels necrotising vasculitis. Clinical manifestations are heterogeneous, but most commonly affect the upper and lower respiratory tract and the kidneys. Other organs and systems, such as the skin, eye, gastrointestinal tract, cardiovascular, musculoskeletal, central and peripheral nervous systems may also be involved [1,2].

One of the key features of GPA is necrotising granulomatous inflammation, which can be used in the differential diagnosis with microscopic polyangiitis (MPA) and other GPA mimickers. Granulomas result from a host immune response against poorly degradable intracellular pathogen or non-degradable extra-cellular material. Histologically, they consist of nodular collections of immune cells, including macrophages and multi-nucleated giant cells, surrounded by T-lymphocytes with variable eosinophils, B-lymphocytes, and plasma cells [3].

Renal abnormalities are observed in 70% of GPA patients and are among the most important disease features due to their prevalence and impact on prognosis. Renal involvement is usually characterised by diffuse pauci-immune crescentic necrotising glomerulonephritis (GN) [4,5]. 

GPA may infrequently occur as a tumour-like lesion, resulting in diagnostic and therapeutic dilemmas. Indeed, such lesions can be difficult to differentiate from other types of infectious, inflammatory, or tumoral masses [6]. The most common sites are the ear, nose, and throat (ENT) district, the breast and the kidney, but GPA-related masses have been described also in the pancreas, eustachian tube, pleura, gingiva, heart, and orbit [3]. 

Herein, we present the results of a systematic literature review, starting from a personal case, with the aim of better defining the characteristics of GPA patients with renal tumour-like lesions, especially in terms of diagnostic work-up, treatment approaches, and outcome.

## 2. Methods

We conducted a systematic literature review to retrieve published cases of GPA patients presenting with renal tumour-like lesions. This systematic review was performed according to the guidelines of the Preferred Reporting Items for Systematic Reviews and Meta-Analyses (PRISMA) [7] and has no PROSPERO registration number.

### 2.1. Information Sources and Search Strategy

A comprehensive search was performed using MEDLINE/PubMed (via OVID), EMBASE (via OVID), and Cochrane Library (via Cochrane Central) databases, from inception of each database to 18 November 2023, using the following terms: “granulomatosis with polyangiitis”; “Wegener”; “Wegener granulomatosis”; “renal mass”; “kidney mass”; “kidney neoplasms”; “renal pseudotumor or carcinoma”; “renal cell”; “tumour-like”.

### 2.2. Inclusion and Exclusion Criteria

The search included randomised controlled trials, observational studies, case series, and case reports with a sample size of one or more patients, without any language restrictions. Editorials, narrative reviews, conference abstracts, comments, and secondary articles were excluded. The inclusion criteria consisted of case reports or case series featuring patients diagnosed with renal masses associated with GPA, whether occurring as isolated or within the context of a systemic disease. No restrictions were made regarding age, gender, sex, race/ethnicity, or nationality. Divergences were resolved by consensus. Two authors (L.I. and R.P.) independently reviewed the articles to determine their suitability based on the inclusion/exclusion criteria, followed by manual searches and citation matching for potential additional articles. Publications were screened using a 2-step approach. First, titles and abstracts were screened to identify potentially eligible studies. Then, the full text of these articles was reviewed. Discrepancies were resolved by adjudication by a third reviewer (A.D.). All authors approved the final selection.

### 2.3. Data Extraction

Data extracted, when available, for both our case and those retrieved from the literature, included information on sex, age, extra-renal organ involvement, acute phase reactants (CRP and ESR), kidney laboratory tests, imaging, location of the renal mass, biopsy, histological lesions (presence of granuloma, glomerulonephritis, vasculitis, and/or fibrosis), exclusion of other causes of granuloma, treatment (glucocorticoid, immunosuppressive induction, and maintenance therapy), surgery treatment (nephrectomy), outcomes, and imaging evaluation in response to therapy. Meta-data from each study, such as lead author name and publication year, were also listed. One author (R.P.) reviewed the entire data extraction process.

### 2.4. Risk of Bias Assessment

Risk of bias was independently evaluated by two authors (L.I. and R.P.). The quality of the studies was assessed using Murad’s tool for case series and case reports [8]. This approach comprises five questions: (1) Does the patient(s) represent(s) the whole experience of the investigator/centre or is the selection method unclear to the extent that other patients with similar presentation may not have been reported?, (2) was the diagnosis accurately established, (3) were other significant diagnoses ruled out, (4) were all crucial data referenced in the report, and (5) was the outcome correctly determined? Each query received a score of 1 (indicating yes) or 0 (indicating no). The study’s quality was categorised as high, moderate, or low based on total scores of 5, 4, or ≤ 3, respectively. Discrepancies were resolved by a third author (A.D.).

## 3. Results

### 3.1. Personal Case Report

The case report in this study is presented according to the CARE criteria [9].

A 49-year-old Caucasian female was admitted to the Rheumatology Unit of Padua University Hospital (Padua, Italy) due to persistent high fever, generalised weakness, and arthro-myalgias. Previously, she had suffered from Human Papilloma Virus (HPV) infection and anxiety disorder. In addition, she reported a smoking habit, consuming half a pack of cigarettes a day for approximately 30 years. 

Blood tests revealed an elevated C-reactive protein (CRP, 161 mg/L) and erythrocyte sedimentation rate (ESR, 120 mm/h), and anaemia (Haemoglobin, 104 g/L). Kidney function tests and urinalysis were normal. Additional laboratory tests (serum angiotensin converting enzyme, β2-microglobulin, rheumatoid factor, complement fractions, immunoglobulin G4, vitamin D, serum calcium) and infectious disease screenings (blood and urine cultures, Treponema Pallidum tests, tuberculosis assay, serology for cytomegalovirus, Epstein–Barr virus, and hepatitis viruses) were unremarkable or negative. Abdominal ultrasound (US) showed normal renal echo-structure, with a single small echogenic nodule and a cyst of the left kidney. Given the history of persistent fever of unknown origin, an ^18^F-fluorodeoxyglucose positron-emission tomography/magnetic resonance (^18^F-FDG-PET/MR) scan was performed, showing several hypo-intense, hypo-vascularised rounded areas on T2-weighted images in both kidneys. These areas showed an intense accumulation of the ^18^F-FDG, with a maximum standardised uptake value (SUV) of 20 (Figure 1).

The preference for ^18^F-FDG-PET/MR was due to its availability at our centre and its notable spatial resolution, along with significantly lower radiation exposure compared to PET-computed tomography. 

None of the masses were gadolinium-enhanced, although two of them exhibited a thin peripheral ring, resembling a perinephric abscess. A contrast-enhanced computer tomography (CT) scan confirmed the presence of several bilateral renal hypovascular nodular areas. These areas measured up to 25 mm in the right and 20 mm in the left kidney (Figure 2). 

Based on ^18^F-FDG-PET/MR and CT scan findings, a renal lymphoproliferative lesion was suspected and a renal core needle biopsy was performed. Histological examination showed a mixed inflammatory infiltrate with numerous lymphocytes, plasma cells, neutrophils, eosinophils, and nodular collections of histiocytes, with occasional multinucleated giant cells and foci of necrosis, consistent with necrotising granulomas. This infiltrate was associated with areas of sclerosis. The periodic acid-Schiff (PAS) and Zhiel–Neelsen stains were negative and the immunohistochemical analysis did not show any increase in immunoglobulin G4-positive plasma cells (Figure 3). Further serological tests disclosed positive anti-proteinase 3 (PR3) anti-neutrophil cytoplasmic antibody (ANCA) at a titre of 42.3 KU/L. This finding and the evidence of necrotising granulomas at kidney biopsy subsequently confirmed the diagnosis of GPA. 

The patient was treated with high-doses of glucocorticoids (40 mg of methylprednisolone daily) and rituximab (RTX), 1000 mg twice, two weeks apart. Prednisone was gradually tapered over 6 months to a dose of 2.5 mg/die and was discontinued after 8 months. Seven months later, a follow-up CT scan confirmed a significant decrease in size of the largest renal lesions and the complete regression of the others. Clinical and laboratory improvements were also observed. For remission maintenance, 500 mg of RTX every 6 months was administered in the following 18 months. 

### 3.2. Characteristics of the Included Studies

The preliminary search produced 143 papers, with 36 articles being duplicates. Additionally, 60 articles were excluded based on title and abstract screening. Eligibility assessment was conducted on 47 articles. Out of these, 17 were excluded as they did not meet the inclusion criteria (reviews, editorials, conference papers), while 11 were added after manual searches and citation matching. 

Finally, data were extracted from 41 articles, resulting in a total of 42 GPA patients presenting with renal masses: 23 (54.8%) were men and the median age was 45.3 (29.8–59.8) years [10,11,12,13,14,15,16,17,18,19,20,21,22,23,24,25,26,27,28,29,30,31,32,33,34,35,36,37,38,39,40,41,42,43,44,45,46,47,48,49,50]. 

The PRISMA flow chart of the results of study selection is shown in Figure 4.

### 3.3. Risk of Bias of Included Studies

Out of the 41 papers included, 11 (26.8%) showed a risk of bias in two or more domains, resulting in low-quality studies, while the remaining ones were judged to be of moderate quality. In papers of moderate quality, imaging, histology, and serology data were always reported. A consistent documentation of patient outcomes was noted in every case. All studies exhibited a risk of bias in the selection domain, due to concerns about underreporting. 

A detailed description of the risk of bias in each domain among the studies is provided in Appendix A.

### 3.4. Synthesis of the Results

The main features of the patients included in this review are summarised in Table 1 and Appendix A. 

Tumour-like lesions were bilateral in 10 cases (23.8%), and unilateral in the other 32 patients (76.2%). Notably, three patients presented with multiple recurrent unilateral masses. Isolated renal lesions were observed in four (9.5%) individuals, while in most cases, renal masses occurred with other concomitant organ involvement, mostly represented by ENT (66.7%) and pulmonary (38.1%) manifestations.

An increase in acute phase reactants was reported in all cases where available, and it was associated with kidney abnormalities in approximately half of the patients (22 out of 42, 52.4%). Positive ANCA was found in 32 (86.5%) cases (data available in 37 patients), and anti-PR3 in 24 (75%) of them. Two anti-PR3 positive patients exhibited only cytoplasmic staining pattern by indirect immunofluorescence.

In almost all cases (95.2%), the diagnosis of GPA was confirmed by histological examination, except in two cases where the diagnosis was only based on clinical and serological features. Diagnostic tests included US in 19 (45.2%) cases, CT scan in 33 (78.6%), magnetic resonance imaging (MRI) in 6 (14.3%), and PET scan in 5 (11.9%).

Kidney biopsy was performed in 27 (64.3%) patients, unfortunately, the biopsy was non-diagnostic in 2 patients [18,32]. Indeed, a re-biopsy was performed in the two cases to achieve a definite diagnosis [33,34]. Nephrectomy was performed in 19 (45.2%) patients, partial in 5 (26.3%), and radical in 14 (73.7%). Histological examination of both kidney biopsy and nephrectomy (data available in 39 cases) showed the presence of granulomatous inflammation in 35 (89.7%) specimens. Glomerulonephritis was less frequent, found in 26 (66.7%) specimens. Therefore, isolated extravascular granuloma without any other histological changes was confirmed in seven (17.9%) cases.

Clinical, laboratory, and imaging improvements were achieved using different treatment approaches. Glucocorticoids were used in 35 (83.3%) patients, and immunosuppressive agents were used in 34 (81%): cyclophosphamide in 21 (61.8%), rituximab in 10 (29.4%), methotrexate in 3 (8.8%), and azathioprine in 2 (5.9%). Notably, two patients were treated with a combination of rituximab and cyclophosphamide. Overall, 36 patients (87.8%, data available in 41 cases) achieved complete remission after first-line treatment. There were four primary failures with cyclophosphamide, and one of them subsequently responded to rituximab. Another patient achieved remission after radical nephrectomy, following the failure of cyclophosphamide [20].

Four patients experienced a poor prognosis as they developed end-stage kidney disease and severe systemic involvement with organ-threatening manifestations. Notably, all these patients, who had an unfavourable prognosis, did not receive immunosuppressive treatment. One patient died from massive haemoptysis and cardiopulmonary arrest before the initiation of induction treatment, and another one died due to severe central nervous system complications after the failure of cyclophosphamide [16,25].

Interestingly, all seven patients who exhibited isolated granulomatous inflammation without any other histological changes in kidney had a favourable prognosis.

## 4. Discussion

### 4.1. Epidemiology of the GPA Renal Masses

In the present systematic review, we identified 42 cases of patients affected with GPA, presenting with tumour-like lesions of the kidney. Additionally, we reported an exceptionally rare case, seen at our Unit, characterised by isolated, multiple bilateral renal tumour-like lesions, without extra-renal manifestations. Isolated renal mass lesions, without any other organ manifestations, were extremely rare and observed in only four cases reported in the literature [20,31,34,36].

Inflammatory tumour-like lesions are uncommon and distinct manifestations of GPA. Goulart et al. used the term “tumefaction” to describe such manifestations and their tendency to lead to tissue destruction [3,34]. Tumour-like lesions were reported in the respiratory tract, the lungs, and the skin and rarely in other districts, such as breast, ocular system, central and peripheral nervous system, gastro-intestinal tract, urogenital system, kidneys, and retro-peritoneum [3].

Notably, there were two case reports of a single renal mass, associated with glomerulonephritis, classified as both immunoglobulin G4-related disease (IgG4-RD) and GPA. The close link and pathogenetic similarities between IgG4-RD and GPA were extensively reported. ANCA predominantly belongs to the IgG1 and IgG4 subclasses and, like IgG4-RD, their generation may occur following prolonged or repetitive exposure to antigens in the context of a type 2 immune response [41,43,51].

Tumour-like lesions are not exclusive to GPA. They have also been observed in other systemic vasculitides including giant cell arteritis, Takayasu arteritis, polyarteritis nodosa, Behçet’s disease, and primary central nervous system vasculitis, with those related to GPA being the most common [52,53,54].

Moreover, several reports in the literature have described the simultaneous occurrence of GPA and malignancies [42,55,56,57,58]. A case-control retrospective study found a numerically, but non significantly, higher prevalence of malignancies in GPA patients, than in rheumatoid arthritis (RA) individuals. The most common cancer in GPA was renal cell carcinoma, with a prevalence of 34.8% compared to 5.6% in RA patients (*p* = 0.0464) [59]. Despite these data, a subsequent case-control study, based on a cohort of 293 GPA patients and 1:10 age- and sex-matched controls, did not confirm a clear association between cancer and GPA onset [60].

### 4.2. Pathophysiology of Granuloma

Extravascular granulomatous inflammation, often associated with PR3-ANCA autoantibodies, distinguishes GPA from MPA. The classical histological triad includes granulomatous inflammation, geographic necrosis, and necrotising small vessel vasculitis [61].

In GPA, a combination of genetic and environmental factors serve as initial triggers for the recruitment and activation of neutrophils [3]. Extravascular primed neutrophils infiltrate the tissues leading to the formation of neutrophil-rich micro-abscesses, which represent an early phase of granuloma formation. This process involves overfed macrophages and multinucleated giant cells, contributing to tumour-like lesions. The granuloma can occlude blood vessels, fostering tissue damage, potentially infiltrating surrounding tissues, leading to necrosis and organ damage [3,62].

Furthermore, in the development of tumour-like mass lesions, Negi et al. proposed that the mass consists of edema surrounding vessel inflammation, while Boubenider et al. hypothesised that ischemic changes resulting from vasculitis induce hypertrophy and cystic alterations of the renal parenchyma, prompting a hypertrophic appearance on imaging studies [15,28]. In addition, follicle-like structures consisting of T- and B-cells are promoted by the release of pro-inflammatory cytokines from the surrounding cells [3,62]. This contains autoreactive B-cells that produce PR3-ANCA [62,63]. T- and B-cell compartments are driven also by various environmental factors including infectious agents (such as simultaneous infection of cytomegalovirus and Epstein–Barr virus or Shiga-like toxin produced by Staphylococcus aureus), drugs (such as propylthiouracil, hydralazine, and cocaine), UV radiation deficiency, and silica dust. Environmental, genetic, and epigenetic factors interact in a complex manner to promote T-cell persistence and the development of autoreactive B-cells into ANCA-producing plasma cells [63,64,65,66].

On the other hand, sustained interleukin 6 (IL-6) expression and PR3 antigen binding on apoptotic neutrophils, result in the stimulation of monocytes and promote giant cell and granuloma formation. In keeping with this, a recent study clearly demonstrated that peripheral blood mononuclear cells stimulated by PR3 antigen can form granuloma-like structures in an IL-6 dependent manner. Moreover, in a granuloma animal model, using zebrafish embryos and larvae, PR3-driven granulomatous reactions were halted by IL-6 blocking, thus providing a rationale for novel therapeutic approaches [66,67].

### 4.3. Clinical Presentation and Diagnosis

Renal masses can be the only organ abnormalities in GPA, but they are extremely rare. The tumour-like lesions frequently co-occur with constitutional signs and symptoms, such as weight loss, fatigue, fever, and sweats. When not isolated, they are associated with ENT involvement and less frequently with pulmonary, peripheral nervous system, central nervous system, skin, or ocular manifestations. Hence, urinary symptoms are uncommon and significant changes in renal function tests are observed in only half of the cases.

There are no diagnostic criteria for GPA; therefore, the definite diagnosis relies on a comprehensive evaluation encompassing clinical assessment, serological testing for ANCA, imaging findings and, when available, histological features.

In patients presenting with renal masses, several disorders should be ruled out before posing a diagnosis of GPA-related tumour-like lesion. This includes primary renal cancers, kidney metastasis from extra-renal solid neoplasms, lymphomas, tuberculosis, xanthogranulomatous pyelonephritis, septic abscesses, malakoplakia, IgG4-RD, Erdheim–Chester disease, or other inflammatory pseudo-tumours.

While ANCA serves as a sensitive and specific tool to support the diagnosis of ANCA-associated vasculitis, it is essential not to rely solely on ANCA serology for the diagnosis. These autoantibodies can be present in other inflammatory diseases, infections, or may be induced by specific drugs. On the other hand, the absence of ANCA does not rule out an ANCA-associated vasculitis [41,42,43,55,56,57].

Imaging is useful in the characterisation of masses but lacks the specificity required for a definitive differential diagnosis. However, the imaging features of GPA tumour-like masses in other organs, such as the lungs, closely resemble those found in the kidneys [68].

At US examination, the kidneys appear enlarged with normal corticomedullary differentiation. GPA renal tumour-like lesions are described as non-vascular hypoechoic, hyperechoic, or heterogeneous nodules, and may occasionally feature a hyperechoic rim. US is highly sensitive in detecting GPA-related renal lesions and represents an effective screening tool in these cases. CT scans reveal hypodense or isodense masses, which can be irregular and exophytic. They exhibit minimal or absent enhancement, but a thin contrast enhancement cortical rim can be observed. Frequently, GPA-related renal tumour-like lesions have unclear margins, and both infiltrating and non-infiltrating masses have been documented. Enlarged lymphnodes at the renal hilum or in the para-aortic region can also be observed.

Data on the MRI imaging of these lesions are very limited. They appear isointense in T1 as compared to the surrounding renal parenchyma and hypointense with a hyperintense centre in T2-weighted sequences. The hyperintense centre could indicate central necrosis. The masses are hypovascular and hypoenhancing on contrast scans and exhibit a variable degree of restriction of diffusivity, including cases where apparent diffusion coefficient (ADC) map is centrally hyperintense with a hypointense rim. Finally, these lesions exhibit significant ^18^F-FDG metabolism on PET scans. This functional imaging technique can be used for diagnostic screening in known GPA patients to identify additional lesions or to assess the treatment response.

Differentiating pseudo-tumours from malignant lesions such as renal cell carcinoma, lymphoma, or other potential inflammatory cases can be challenging. However, some features can hint to the correct diagnosis. T2 hypointensity and poor diffusion restriction can aid in differentiation from renal cell carcinoma and lymphoma, while absent enhancement can be useful in excluding abscess. Differential diagnosis with other inflammatory tumour-like masses, such as IgG4-RD or Erdheim–Chester disease, requires histological examination. Indeed, histological assessment remains essential to achieve a definitive diagnosis. The GPA renal masses are usually characterised by granulomatous inflammation, followed in frequency by glomerulonephritis, vasculitis, necrosis, fibrosis, and glomerulosclerosis. Conversely, tubular involvement is extremely rare.

To characterise a renal mass, the least invasive procedure is a CT- or US-guided kidney biopsy. Biopsy enables assessment for the presence of granulomatous inflammation, fibrosis, vasculitis, or pauci-immune glomerulonephritis. It also facilitates diagnosis in cases where specific clinical features of GPA are absent and ANCA tests yield negative results. Moreover, if radiological features do not clearly indicate a malignancy, biopsy offers patients an alternative to undergoing highly invasive surgery, thereby preserving healthy kidney tissue. However, performing a kidney biopsy is not recommended in patients with a high suspicion of renal cell carcinoma. In highly uncertain cases, nephron-sparing surgery should be preferred over biopsy and radical nephrectomy.

### 4.4. Treatment and Outcome

In both the recent American College of Rheumatology/Vasculitis Foundation (ACR/VF) 2021 and European League Against Rheumatism (EULAR 2022) recommendations, granulomatous manifestations are not considered as severe disease manifestations of GPA, while glomerulonephritis is included among organ life-threatening manifestations [69,70].

Half of the patients (*n* = 21, 50%) gathered from our literature review were treated with cyclophosphamide, resulting in four primary failures. On the other hand, those treated with rituximab, azathioprine or methotrexate achieved disease remission. These data must be interpreted with caution due to possible biases related to the heterogeneity of patients included and to the variety of clinical manifestations. The choice of treatment for granulomatous manifestations remains a subject of debate. Rituximab may be preferred due to its superior safety profile compared to cyclophosphamide, particularly in ANCA PR3-positive patients [71]. Some real-life data also suggest its efficacy in localised granulomatous manifestations, although treatment response is often slow [72]. It is unclear whether these lesions represent residual fibrotic scars from previous inflammatory processes. Supporting this theory, Starr et al. demonstrated two distinct patterns in orbital mass biopsies from seven GPA patients treated with rituximab: one characterised by significant fibrosis and lymphocytic depletion, and the other by acute inflammation and vasculitis, indicating active disease despite treatment [73]. These morphological findings may inform treatment strategies; specifically, patients with significant fibrosis and minimal inflammation might benefit more from debulking surgery.

Regardless of the treatment regimen, an early diagnosis and timely initiation of immunosuppressive therapy can prevent disease progression and irreversible damage. Indeed, it is important to emphasise that the growth of these masses can be very rapid [25,35].

No renal relapses were observed in patients reported in the present literature review, but the short follow-up prevents the drawing of definitive conclusions. Only 7.1% of patients experienced one or two systemic GPA disease flares. The relapse rates appear significantly lower when compared to those of historical cohorts, in which 13.7–44% of patients experienced a relapse at 18–36 months [74].

In our review, only two patients (4.7%) with multiple severe organ manifestations died. By contrast, no deaths were observed among the patients with isolated renal masses.

These findings suggest that patients with isolated kidney granulomatous inflammation exhibit a less severe disease phenotype with an overall good prognosis.

## 5. Conclusions

Our results suggest that renal tumour-like lesions are rare manifestations of GPA and emphasise the importance of considering this disease in the differential diagnosis of kidney masses.

Laboratory and imaging findings are crucial for the diagnosis of GPA-related tumour-like lesions, but they are non-specific. Renal biopsy plays a significant role in the diagnostic work-up, being less invasive than surgery, highly informative in most cases, and can prevent unnecessary nephrectomy while driving an appropriate medical therapy. Isolated kidney masses do not appear to result in unfavourable outcomes, as long as patients receive early diagnosis and timely treatment.

Future research should prioritise the identification and validation of novel biomarkers that can facilitate early detection and accurate diagnosis of GPA in tumour-like masses. These biomarkers should ideally exhibit high sensitivity and specificity, enabling clinicians to distinguish GPA-related masses from other pathologies promptly and reliably. Moreover, prospective studies with larger and more diverse patient cohorts are imperative to comprehensively assess the safety and efficacy of existing treatment modalities for GPA-associated tumour-like masses.

## Figures and Tables

**Figure 1 diagnostics-14-00566-f001:**
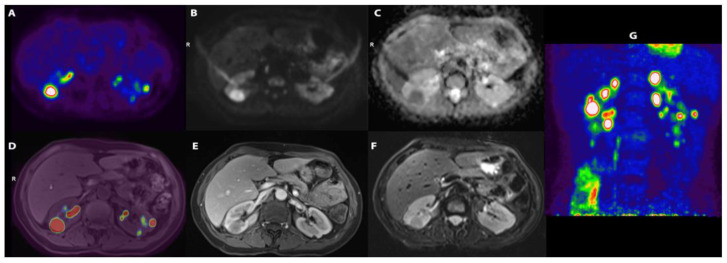
(**A**) [^18^F]-FDG transaxial image demonstrating a number of areas of uptake of the tracer; (**B**) DWI-b1000 (high. Res. Resolve) and (**C**) transaxial DWI-ADC (high. Res. Resolve) showing corresponding areas of restriction of diffusivity; (**D**) [^18^F]-FDG and T1 vibe fused images; (**E**) transaxial T1 vibe after contrast enhancement not showing significative enhancement; (**F**) transaxial T2 TSE; (**G**) [^18^F]-FDG maximum intensity projection (MIP) showing a number of lesions with high uptake of the tracers in kidneys.

**Figure 2 diagnostics-14-00566-f002:**
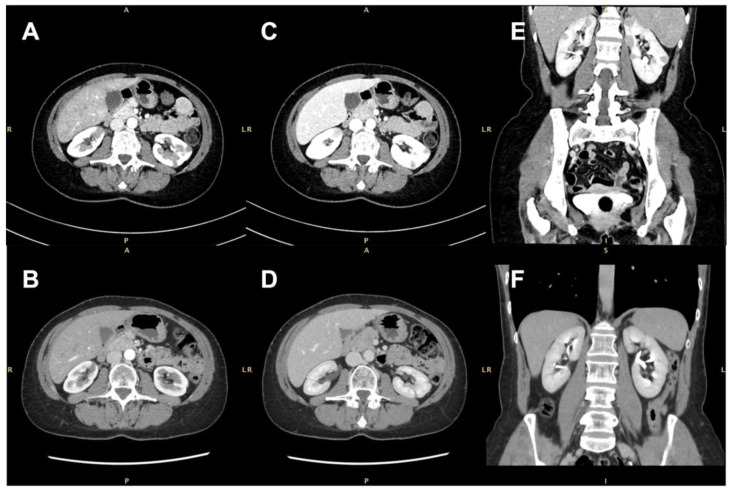
Contrast-enhanced computed tomographic (CT) scan showing several bilateral renal hypovascular nodular areas (arrows). (**A**,**B**) Axial arterial phase at diagnosis and 7 months after treatment with rituximab; (**C**,**D**) axial early-venous phase at diagnosis and 7 months after treatment with rituximab; (**E**,**F**) coronal portal-venous phase at diagnosis and 7 months after rituximab therapy.

**Figure 3 diagnostics-14-00566-f003:**
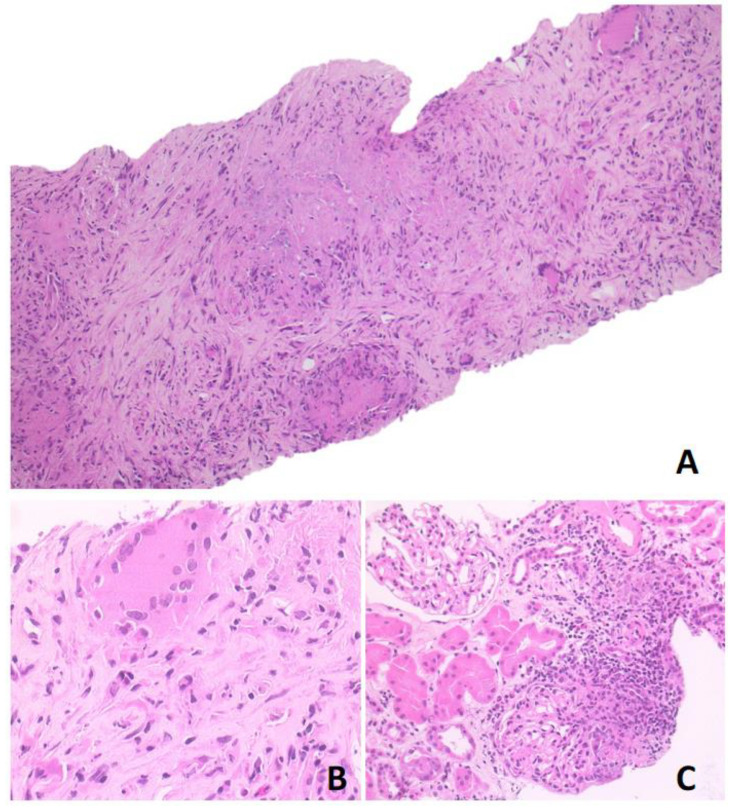
Photomicrographs in hematoxylin-eosin stained section. (**A**) Original magnification ×10 showing granulomatous nodules with necrosis and sclerosis; (**B**) original magnification ×40 showing granulomatous structure; (**C**) original magnification ×20 showing neutrophilic and eosinophilic infiltrate, with multinucleated giant cells. The surrounding renal parenchyma was spared.

**Figure 4 diagnostics-14-00566-f004:**
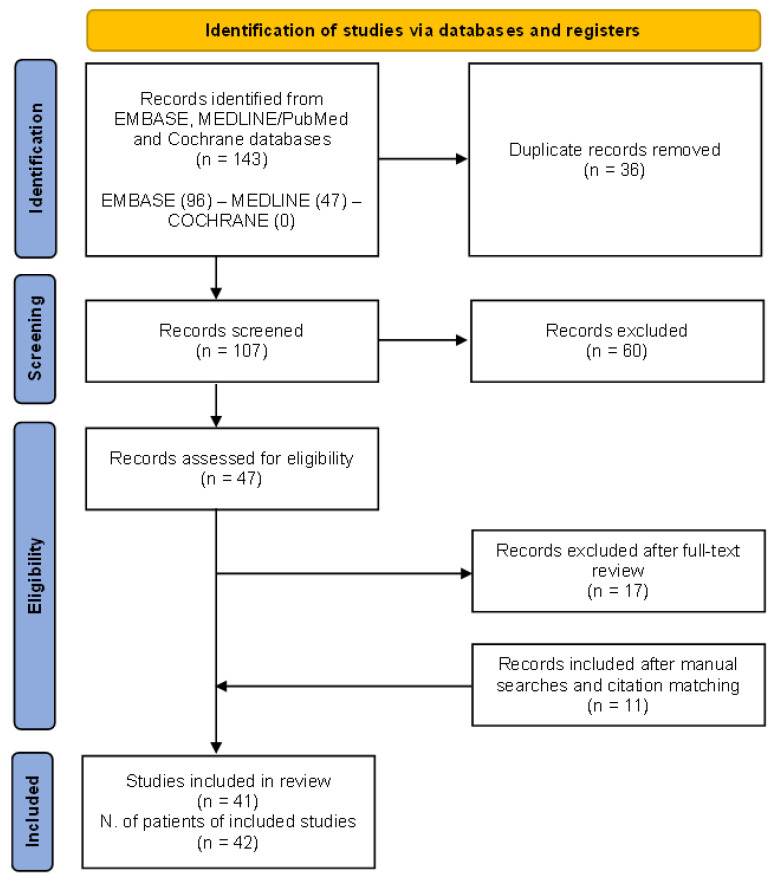
The (Preferred Reporting Items for Systematic Reviews and Meta-Analyses) PRISMA 2020 flow diagram for the systematic review performed on MEDLINE/PubMed, EMBASE, and Cochrane databases to retrieve cases of granulomatosis with polyangiitis patients presenting with renal tumour-like lesions.

**Table 1 diagnostics-14-00566-t001:** Clinical and demographic features of the 42 GPA patients with renal masses retrieved from the current literature and the present case.

Author	Sex/Age (y)	Extra-Renal Organ Involvement	ANCA	Kidney Tests	Imaging	Localisation	Biopsy	Kidney Removal	Histology	Treatment	Outcome
Tiwari [10]	F/60	ENT, eye	MPO	Normal	US, CECT, PET	Unilateral	Yes	No	Granuloma	MTX	Remission
Maguire [11]	F/27	ENT, lung	PR3	NA	Renal arteriography	Unilateral	NA	Partial	Granuloma/Vasculitis	CYC	Remission
Schapira [12]	M/45	ENT, lung	NA	UP	Gallium scan, US	Unilateral	No	Partial	Granuloma/Vasculitis	CYC	Remission
Schydlowsky [13]	M/47	ENT, lung	Positive	Normal	US, CECT	Unilateral	No	Radical	Granuloma/Vasculitis	CYC	Remission
Smith [14]	F/52	Eye	NA	WBCs	US, CT	Unilateral	No	Radical	Granuloma/Vasculitis	AZA	Remission
Boubenider [15]	F/45	ENT, skin	PR3	Renal failure, RBCs, UP	US, CT	Unilateral	Yes	Radical	Granuloma/Vasculitis	None	ESRD
Fairbanks [16]	M/68	ENT, lung	MPO	Normal	CECT	Unilateral	Yes	No	Granuloma/Vasculitis	MTX	Remission
Dufour [17]	M/70	ENT, PNS, lung	MPO	Renal failure	CT	Unilateral	No	Radical	Granuloma/Vasculitis	CYC	Relapsing disease
Dufour [17]	M/67	MSK, lung, CNS	PR3	Renal failure	CT	Unilateral	NA	No	NA	CYC	Death
Thomas [18]	F/48	ENT, PNS, eye	NA	Renal failure	IVU, CT	Unilateral	Yes	Radical	Granuloma/Vasculitis	CYC	Remission
Verswijvel [19]	M/24	ENT, spleen	PR3	Renal failure, RBCs	US, CECT, MRI	Unilateral	Yes	No	Vasculitis	CYC	Remission
Carazo [20]	M/29	Eye	PR3	Normal	US, IVU, CECT	Bilateral	Yes	Radical	Granuloma/Vasculitis	CYC	Relapsing disease
Kapoor [21]	M/22	-	PR3	Renal failure, RBCs, UP	US, MRI	Bilateral	Yes	No	Vasculitis	-	ESRD
Leung [22]	M/66	ENT, skin	MPO	Normal	US, CT	Bilateral	Yes	No	Granuloma	MTX	Remission
D’Hauwe [23]	F/14	ENT, MSK	Negative	WBCs	US, CECT	Unilateral	Yes	No	Granuloma	RTX+MTX	Remission
Krambeck [24]	M/61	ENT, CNS	Negative	Normal	CECT	Unilateral	No	Partial	Granuloma	AZA	Remission
Vandergheynst [25]	M/32	PNS	PR3	UP	CT	Unilateral	Yes	Partial	Granuloma/Vasculitis	CYC	Remission
Vandergheynst [26]	F/23	ENT, skin, endocrine system	MPO	Normal	PET, CEUS	Bilateral	Yes	No	Granuloma/Vasculitis	RTX	Remission
Sichani [27]	F/22	ENT, lung, DAH	Positive	UP, RBCs	US, CECT	Unilateral	Yes	No	Granuloma/Vasculitis	-	Death
Negi [28]	M/40	ENT	PR3	Renal failure	CT	Bilateral	No	No	NA	NA	Remission
Lo Gullo [29]	M/38	ENT, lung	PR3	Normal	US, CECT, PET	Unilateral	Yes	No	Granuloma	RTX	Remission
Frigui [30]	F/59	ENT, eye	PR3	UP	US, CECT	Bilateral	Yes	No	Granuloma/Vasculitis	CYC	Remission
Xu [31]	M/55	-	PR3	RBCs	CT	Unilateral	No	Radical	Vasculitis	NA	Remission
Roussou [32]	F/72	ENT	MPO	Normal	CECT, MRI	Unilateral	No	Radical	Granuloma/Vasculitis	CYC	Remission
Ahmed [33]	F/28	Lung	PR3	RBCs	US, CT	Bilateral	Yes	No	Granuloma/Vasculitis	CYC	Remission
Ward [34]	F/48	ENT, CNS, lung	PR3	NA	CT	Unilateral	Yes	Radical	Granuloma/Vasculitis	CYC+RTX	Remission
Yamamoto [35]	M/60	-	MPO	Normal	CECT, MRI	Unilateral	Yes	Radical	Granuloma/Vasculitis	-	Remission
Fu [36]	F/33	ENT	Negative	Normal	US, PET	Bilateral	Yes	No	Granuloma/Vasculitis	CYC	Remission
Higashihara [37]	F/75	-	Negative	Renal failure	CECT	Unilateral	No	Radical	Granuloma/Vasculitis	-	Remission
Dai [38]	M/32	ENT, lung	PR3	Normal	MRI, PET	Unilateral	Yes	No	Granuloma	CYC	Remission
Guo [39]	F/71	Lung, eye	PR3	Normal	CT	Bilateral	Yes	No	Granuloma	RTX	Remission
Kumar [40]	F/27	ENT	PR3	Normal	US, CECT, IVU	Unilateral	Yes	No	Granuloma	RTX	Remission
Reeders [41]	M/46	Lung, skin, MSK	PR3	Renal failure	CT	Unilateral	No	Radical	Granuloma/Vasculitis	RTX+CYC	Remission
Villa-Forte [42]	M/45	ENT, lung	NA	NA	NA	Unilateral	No	Radical	Granuloma/Vasculitis	CYC	ESRD
Boncoraglio [43]	M/47	ENT	PR3	Renal failure, RBCs, UP	US, CT, MRI	Unilateral	No	Partial	Granuloma/Vasculitis	RTX	Remission
Abudaff [44]	M/20	ENT	PR3	Renal failure, RBCs, UP	US, CT	Unilateral	Yes	No	Vasculitis	RTX	Remission
Bicakcigil [45]	F/47	ENT, breast, spleen	NA	NA	CT	Bilateral	Yes	No	NA	CYC	Remission
Gregorini [46]	M/46	ENT, CNS	PR3	Normal	CT	Unilateral	Yes	No	Granuloma	RTX+CYC	Remission
Kaikoi [47]	F/76	ENT, eye	PR3	Minimal UP	CECT	Unilateral	Yes	No	Granuloma/Vasculitis	CYC	Remission
Nketiah Sarpong [48]	M/57	Lung, DAH	PR3	Renal failure, RBCs, UP	NA	Unilateral	Yes	No	Granuloma/Vasculitis	RTX	Remission
Ramasamy [49]	F/39	NA	Negative	Renal failure	NA	Unilateral	No	Radical	Granuloma/Vasculitis	NA	ESRD
Varkala [50]	M/22	Lung, DAH, heart, skin	PR3	Renal failure, RBCs, WBCs, granular casts	CECT	Unilateral	Yes	No	Vasculitis	CYC	Remission
Present case	F/49	-	PR3	Normal	US, CECT, PET-MR	Bilateral	Yes	No	Granuloma	RTX	Remission

ANCA: antineutrophil cytoplasmic antibodies; AZA: azathioprine; CECT: contrast enhanced computed tomography; CEUS: contrast-enhanced ultrasonography; CNS: central nervous system; CT: computer tomography; CYC: cyclophosphamide; DAH: diffuse alveolar haemorrhage; ENT: ear, nose, and throat; ESRD: end stage renal disease; IVU: intravenous urography; MPO: myeloperoxidase; MRI: magnetic resonance imaging; MTX: methotrexate; NA: not available; PET: positron-emission tomography; PNS: peripheral nervous system; PR3: proteinase 3; RBCs: red blood cells; RTX: rituximab; UP: urine protein; US: ultrasound; WBCs: white blood cells.

## Data Availability

The de-identified data of the case report are available on request from the corresponding author. All data for the systematic review are available within the article.

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
