# Peer review of "Cutting-Edge Strategies for Renal Tumour-like Lesions in Granulomatosis with Polyangiitis: A Systematic Review"

_diagnostics, 2024, doi:10.3390/diagnostics14050566_

Round 1

Reviewer 1 Report

Comments and Suggestions for Authors

The manuscript reports intersting case report and systematic review regarding Tumor -like Lesions in Granulomatosis with Polyangiitis (GPA).

I have some minor considerations regarding your case report.

In your  diagnostic algorithm you initially performed  PET/MR  instead CT scan which is also sensitive (and more easily accessible)  method for detection of eventually malignant lesions and also for detection of lung manifestations of GPA.
Did you evaluate the presence of granuloma related to GPA in trachea and  bronhi with invasive diagnostic methods?

Could you comment some more about your first therapeutic choice ? Your first choice for the treatment was rituximab, and the majority of other authors cited in your manuscript used cyclophosphamide. I suggest that you comment some more about this approach in discussion.
In disussion and conclusion you should  emphasize  that detection of tumor like lesion  related to GPA can avoid nepfrectomy in suspected malignant cases.

Reviewer 2 Report

Comments and Suggestions for Authors

In this review, Iorio et al conducted a systematic literature review on MEDLINE/PubMed, EMBASE and Cochrane databases. Data gathered from the literature were analysed to summarize the diagnostic approach, management, and outcome of GPA-related tumour-like lesions. Literature search yielded 41 articles, concerning 42 GPA patients with renal masses, presenting bilaterally in 23.8% of the cases. Positive PR3-ANCA was observed in 86.5% of the cases. Half of 42 patients showed kidney abnormalities. Treatment with glucocorticoids (83.3%) and immunosuppressive agents (80.9%) resulted in an overall good remission rate and favourable prognosis. They concluded that GPA should be considered in the differential diagnoses of kidney tumour-like lesions. The diagnosis is challenging, and histological examination greatly contributes to the diagnostic work-up. The contents are systematically and clearly organized and will be useful to many readers. There are no particular corrections to be made.

Reviewer 3 Report

Comments and Suggestions for Authors

The manuscript provides a systematic review focused on the rare presentation of tumor-like lesions in Granulomatosis with Polyangiitis (GPA), emphasizing the diagnostic challenges and treatment outcomes associated with renal involvement. It outlines the methodology of a systematic review, presents a personal case, and aggregates findings from the literature to discuss the epidemiology, pathophysiology, clinical presentation, diagnosis, treatment, and outcomes of GPA with renal tumor-like lesions.

Despite the thorough approach, the manuscript could benefit from a broader context regarding the interaction between environmental factors and GPA manifestations, specifically in how varying conditions might modulate disease presentation or outcomes. The study by Al-Awaida et al., 2023, which explores the modulation of wheatgrass toxicity against breast cancer cell lines by simulated microgravity, presents an interesting angle on the impact of environmental factors on disease models, which could enrich the discussion around GPA. Incorporating this reference could provide a nuanced perspective on how similar environmental or experimental conditions might influence the pathophysiology or therapeutic responses in GPA, especially given the focus on renal tumor-like lesions. A suitable place for this reference could be in the discussion section, where the implications of the findings on GPA's broader understanding and management are deliberated. This addition would underscore the importance of considering a wide range of factors in GPA's clinical and therapeutic approaches.

here are some areas where improvements are needed:

  1. Specificity in Methodology: The methodology section, while detailed, lacks specificity in certain areas, such as the criteria for selecting studies for the review. A more rigorous explanation of the inclusion and exclusion criteria, as well as the databases searched, would enhance the reliability of the review.
  2. Data Analysis: The analysis of the collected data seems to be superficial. A deeper statistical analysis, possibly meta-analysis, could provide more meaningful insights into the prevalence, outcomes, and response to treatment of GPA with renal tumor-like lesions.
  3. Discussion on Pathophysiology: The discussion could be enriched by a more detailed exploration of the pathophysiological mechanisms underlying GPA, especially in the context of renal involvement. The current discussion provides a general overview but misses an opportunity to delve into the molecular and cellular mechanisms that contribute to the disease's presentation and progression.
  4. Literature Review: The review of related literature focuses primarily on clinical presentations and outcomes but could benefit from a broader scope that includes recent advancements in diagnostic methodologies and therapeutic interventions. Integrating more current research findings could provide a more comprehensive overview of the subject matter.
  5. Clinical Implications and Future Directions: The conclusion section briefly touches on the implications of the findings for clinical practice but could be expanded to offer more concrete recommendations for clinicians. Additionally, suggesting specific areas for future research would be beneficial to guide subsequent studies in this field.
  6. Reference Utilization: The manuscript could benefit from incorporating a wider range of sources, including recent studies that may offer new perspectives or corroborate the findings presented. For instance, integrating studies that examine the effects of environmental factors on autoimmune diseases could provide additional depth to the discussion.

Overall, while the manuscript provides valuable information on GPA with renal tumor-like lesions, addressing these areas could significantly enhance its contribution to the literature and its utility for both clinicians and researchers.

Comments on the Quality of English Language

The manuscript is generally well-written and demonstrates a good grasp of the English language. However, there are a few areas where improvements could enhance clarity and readability. Some sentences are overly complex, making them difficult to follow. Simplifying these sentences could improve the overall flow of the manuscript. Additionally, there are occasional grammatical errors and inconsistencies in terminology that could be addressed through careful proofreading. Addressing these issues would not only improve the readability of the manuscript but also ensure that the scientific content is communicated effectively
